# Proton pump inhibitor and community pharmacies: Usage profile and factors associated with long-term use

**Lorena Maria Lima de Araújo**[1,2], **Maria Vivyanne de Moura Lopes**[2‡], **Rafael Silva de Arruda**[2‡], **Rand Randall Martins** [1,2]*, **Antonio Gouveia Oliveira**[1,2]

**1** Post-Graduate Program in Pharmaceutical Sciences, Health Sciences Center, Federal University of Rio Grande do Norte, Natal, Rio Grande do Norte, Brazil, **2** Pharmacy Department, Health Sciences Center, Federal University of Rio Grande do Norte, Natal, Rio Grande do Norte, Brazil

☯ These authors contributed equally to this work.
‡ These authors also contributed equally to this work.
* rand.martins@ufrn.br

## Abstract

### Aim

To characterize the usage profile and the factors associated with the prolonged use of proton pump inhibitor drugs in a community pharmacy.

### Methodology

This is a cross-sectional, prospective and observational study involving interviews with 410 patients who acquired PPI for their own use from community pharmacies. To characterize the factors associated with the prolonged use of PPI, a multivariate logistic regression model was used.

### Results

Pantoprazole (42.7%) and omeprazole (31%) were the most acquired PPIs, prescribed mainly by gastroenterologists (49.5%). They are used in the morning, especially for gastrointestinal symptoms, however, they had been consumed for more than 5 years in 30% of cases. The factors associated with prolonged use are old age (OR 1.03 CI95% 1.01–1.05), body mass index (OR 1.07 CI95% 1.01–1.12), use of non-steroidal anti-inflammatories (OR 3.18 CI95% 1.20–8.43) and selective serotonin reuptake inhibitors (OR 3.5 95% CI 1.39–8.88).

### Conclusion

PPIs are adequate in terms of indication and form of use, however, prolonged use associated with old age, being overweight and use of anti-inflammatories and antidepressants is frequent.

**Data Availability Statement:** All relevant data are within the manuscript and Supporting information.

**Funding:** We declare that this study was financed by the Coordination for the Improvement of Higher

Education Personnel - Brazil (CAPES) - Finance Code 001. There was no additional external funding received for this study.

**Competing interests:** The authors have declared that no competing interests exist.

## Introduction

Proton pump inhibitors (PPI) are widely prescribed for the treatment of various dyspeptic diseases such as gastroesophageal reflux, peptic ulcer and gastropathy induced by non-steroidal anti-inflammatory drugs [1]. In 1999, Sweden was the first country to grant omeprazole the status of an over-the-counter (OTC) drug. This decision was soon followed by the European Union, Norway, Switzerland, China, the United States, Canada, Australia, New Zealand and, in Latin America, by Mexico, Colombia and Argentina [2]. In 2013, Esomeprazole was also regulated as OTC throughout the European Union, a measure subsequently followed by Canada and Australia [3]. In Brazil, PPIs are usually marketed in community pharmacies without the requirement of medical prescription, thus resembling a form of OTC dispensation practiced in the vast majority of countries.

OTC drugs are characterized by their use for a short period of time, for a proven indication and with a low risk potential for the patient; medicines related to the digestive system alone were worth $ 4.3 billion in the United States in 2016 [4]. However, there are few studies on the use of OTCs by the general population; little is known about prescription profiles, usage patterns, concomitant medications and proportion of inappropriate use. This also applies to PPIs, especially considering their restricted therapeutic indications, wide use and potential risks associated with their long-term use.

In a retrospective study involving 409 patients admitted to a general hospital, it was found that 76% of PPIs were used inappropriately before admission [5]. Despite the very favorable safety profile that justifies its OTC status, the unsupervised use of PPIs is not without risks. Several studies have reported potential problems related to prolonged use, including respiratory infections, nutritional deficiencies involving decreased absorption of vitamin B12, iron and calcium and even gastric neoplasms [6]. However, some mechanisms are still unclear.

As far as we know, there are no studies in the literature on the use of PPIs sold as OTC. In view of the scarcity of information and the widespread use of these drugs, we conducted a prospective study on the use of PPI based on a random sample in the community. This was done in order to characterize the usage profile and the factors associated with the prolonged use of these drugs in a community pharmacy.

## Methods

### Study design and population

This was an observational and cross-sectional study in patients from a community pharmacy chain that acquired Proton Pump Inhibitors (PPIs) in the city of Natal, Brazil (Jul 2018 to Mar 2020). Individuals over 18 years old who purchased PPIs for their own consumption were included. People with auditory, cognitive, speech and other disorders that made it difficult to collect information were excluded. This study was approved by the Institutional Review Board of the Hospital Onofre Lopes (authorization number 2.446.211) and Informed consent was obtained from all individual participants enrolled in the study.

The sample size was calculated at 385 participants. This quantity ensures, with 95% confidence, a maximum error of the estimates of ± 5 percentage points. The municipality of Natal, located in the Northeast of Brazil, has 800,000 inhabitants distributed among four health districts (east, south, north and west districts). Health district is a geographic area that comprises a population with similar epidemiological and social characteristics, in addition to the health resources to serve it [7]. In the city evaluated, 38% of the population lives in the northern health district, followed by the western (27%), southern (21%) and eastern (14%) districts.

In order to obtain a sample that represented the different segments of the city's population, the selection and recruitment of participants were conducted in community pharmacies from the 4 districts of the city, with the inclusion of a number approximately proportional to the population of each district. Only one community pharmacy chain participated in the research, consisting of 22 establishments distributed throughout the city (3 pharmacies in the north district, 2 in the west, 11 in the south and 6 in the east).

## Data collection

In each pharmacy, a consecutive sample of pharmacy users was made. 'All individuals who acquired PPI were approached about their interest in answering the questionnaire. During the day shift (as 8 a.m. to 4 p.m.), the interviewer remained inside the establishment and approached the participant only after the purchase was completed. The questions were asked orally in a place reserved by the main researcher (pharmacist, LMLA) and auxiliaries (pharmacy students, MVM and RSA). Before the start of data collection, we carried out a pilot study with 10 patients to adjust the questionnaire and train the research team.

The patients were asked about socio-demographic data (age, sex, income, education, smoking, alcohol consumption, weight and height), self-reported diseases and other medications in use (Anatomical Therapeutic Chemical Classification—ATC). PPIs were characterized in relation to the active principle and dose per tablet in mg, time of administration and time of use. Patients were asked if the acquisition of PPI was due to a medical evaluation by a gastroenterologist and made with presentation of the prescription. The reasons for use and knowledge about potential risks of the prolonged use of PPIs were also investigated. Individuals with 3 or more years of frequent use of PPIs were considered to demonstrate prolonged use. This time of use is associated with the appearance of changes in gastrin levels and gastric histopathology, initial factors for the occurrence of several complications of long-term use [8].

## Statistical analysis

Statistical analysis was performed with Stata 15. Data are presented as mean and standard deviation or relative and absolute frequencies when relevant. For the identification of factors associated with prolonged use of PPIs, univariate analysis by logistic regression was used, including the grouping of cases in pharmacies and robust standard errors, with variables with a p-value $<0.10$ included in a multivariate model by logistic regression, considering variables with p $<0.05$ significantly associated. Multicollinearity was tested by calculating variance inflation factors (VIF) for all explanatory variables in the full and the minimal multivariate model. The full model had all values of VIF $< 4.5$ and the minimal model had all values of VIF$< 2.0$, and hence no problems were displayed.

## Results

During the execution of the study, about 520 patients were invited and approximately 20% refused to answer the questionnaire, the main reason for refusal was short time. The study included 410 patients with a predominance of females (62%), with a mean age of 54.1 ± 17.9 years and with high school as the predominant level of education (43.7%). The average body mass index was slightly above the recommended (27.2 ± 4.6 Kg / m2). Regarding the profile of comorbidities, there was a predominance of cardiovascular diseases (42.9%), however, many individuals did not report other chronic diseases (35.6%). Antihypertensive drugs were the most used (23.4%), followed by hypolipidemic drugs (19.5%) and hypoglycemic drugs (12.9%). Table 1 describes the population characterization.

**Table 1. Characteristics of population.**

| Characteristics | Values | | 95%CI | |
| --- | --- | --- | --- | --- |
| Age in years (m, sd) | 54.1 ± 17.9 | | 52.3 | 55.8 |
| Female (n, %) | 254 | 62.0 | | |
| Income by minimum wage * (n, %) | | | | |
| 0–5 minumum wage | 299 | 72.9 | | |
| 6–10 minumum wage | 56 | 13.6 | | |
| Above 10 minimum wage | 50 | 12.4 | | |
| Education (n, %) | | | | |
| Not literate | 15 | 3.7 | | |
| High school | 63 | 15.4 | | |
| Elementary school | 179 | 43.7 | | |
| Higher education | 153 | 37.3 | | |
| Smoker (n, %) | 23 | 5.6 | | |
| Alcohol use (n, %) | 54 | 13.2 | | |
| BMI (m, sd) | 27.2 ± 4.6 | | 26.7 | 27.6 |
| Self-reported diseases (n, %) | | | | |
| Heart problems | 176 | 42.9 | | |
| No other health problems | 146 | 35.6 | | |
| Bone problems | 25 | 6.1 | | |
| Medications in use (m, sd) | 1.9 ± 2.1 | 2.1 | 1.7 | 2.1 |
| Medications ATC class (n, %) | | | | |
| Angiotensin II receptor blockers | 96 | 23.4 | | |
| HMG CoA reductase inhibitors | 80 | 19.5 | | |
| Biguanides | 53 | 12.9 | | |
| No medications | 135 | 32.9 | | |

m, sd: mean and standard desviation; n,%: absolute and relative frequency.

* In 2019, the Brazilian minimum wage per month is $ 998 Reais and is equivalent to $ 257 US Dollar.

Regarding the PPI acquisition profile (Table 2), pantoprazole (40 mg and 20 mg tablets) is the predominant medication (42.7%), followed by omeprazole (40, 20 and 10 mg) with 31%. The use of PPI occurs mainly in the morning, more specifically 15 minutes before breakfast. The occurrence of long-term use (over 5 years) was observed in 27.5% of patients. In addition, we detected a considerable percentage of new users (19.4%) with use for less than 6 months.

Approximately 90% of patients report using PPIs under medical prescription. Among these, the majority were evaluated by gastroenterologists (49.5%), followed by other medical specialists (16.4%) and general practitioners (9.7%). Gastrointestinal discomfort was the most cited motivation for use (78.8%) and the use of PPI motivated by polymedication was 11.1% of the sample. It is important to emphasize that more than 75% of the interviewees do not know the risks of long-term use or cannot describe them, while about 12% of the interviewees believed that prolonged use causes of dementia or cancer. The data is described in Table 3.

The univariate analysis by logistic regression (Table 4) showed that the prolonged use of PPI was related to old age, BMI, the use of non-steroidal anti-inflammatory drugs (NSAIDs) and selective serotonin reuptake inhibitors. The multivariate model maintained the same variables: age (OR 1.03 CI95% 1.01–1.05), higher BMI (OR 1.07 CI95% 1.01–1.12), use of NSAIDs (OR 3.18 CI95% 1.20–8.43) and antidepressants (OR 3.5 CI95% 1.39–8.88). All of which characterize the increased risk of prolonged use of these drugs.

**Table 2. PPI type and administration profile.**

| Characteristics | n | % |
|---|---|---|
| Period of use | | |
| First use | 79 | 19.4 |
| >6 months | 87 | 21.4 |
| 1 to 2 years | 71 | 17.4 |
| >3 years | 58 | 14.3 |
| 5 to 10 years | 112 | 27.5 |
| Time of day of use | | |
| 4h00—8h00 | 355 | 86.6 |
| 8h01—12h00 | 30 | 7.3 |
| 12h01—23h59 | 25 | 6.1 |
| PPI type and dose per tablet (n, %) | | |
| Pantoprazole | | |
| 40 mg | 109 | 26.6 |
| 20 mg | 66 | 16.1 |
| Omeprazole | | |
| 40 mg | 27 | 6.6 |
| 20 mg | 96 | 23.4 |
| 10 mg | 4 | 1.0 |
| Dexlansoprazole | | |
| 60 mg | 30 | 7.3 |
| 30 mg | 9 | 2.2 |
| Esomeprazole | | |
| 40 mg | 35 | 8.5 |
| 20 mg | 20 | 4.9 |
| Lanzoprazole 30 mg | 10 | 2.4 |
| Rabeprazole 20 mg | 1 | 0.2 |

n,%: absolute and relative frequency

## Discussion

In this observational study, from a representative sample of PPI users found in community pharmacies, the main findings were a pattern of use consistent with the indications and dosages expected for PPIs. However, a considerable portion use for an extended period, exceeding more than 5 years of use. In structured interviews, it was observed that most patients report use under medical prescription, with little occurrence of unlicensed indications while following appropriate guidelines for use. However, a third of users have used PPIs for more than 5 years, with long-term use being related to old age, BMI and use of NSAIDs. Regarding the knowledge about potential risks associated with the long-term use of PPIs, most patients do not know or remember.

It is common practice to purchase medicines without a prescription, about 50% of dispensations in community pharmacies in Saudi Arabia are self-medications, of these 27% were for medicines that are not OTC [9]. According to another Saudi study also carried out in community pharmacies, the tendency to increase self-medication is associated with lack of time and difficulties in accessing health services, in addition to financial restrictions as well as the extensive advertising about OTC drugs [10]. In the case of PPIs, self-medication is observed in more than 60% of community pharmacy customers (302 patients from 3 community pharmacies)

**Table 3. Prescription profile of PPIs, motivation for acquisition and knowledge about risks of prolonged use.**

| Characteristics | n | % |
|---|---|---|
| PPI acquired by medical indication | 359 | 87.8 |
| PPI acquired with presentation of medical prescription | 165 | 40.3 |
| Prescriber Specialty | | |
| Gastroenterologist | 193 | 49.5 |
| General practitioner | 38 | 9.7 |
| Other medical specialties | 64 | 16.4 |
| Don't know / Don't remember | 95 | 24.4 |
| Reason for use | | |
| Gastrointestinal symptoms (GI) | 320 | 78.8 |
| Use of medications | 45 | 11.1 |
| Prevention of GI symptoms | 29 | 7.1 |
| Other reasons | 11 | 2.7 |
| Knowledge of prolonged use risk | | |
| Unknown risk | 235 | 57.5 |
| Know, but do not remember | 82 | 20.1 |
| Causes dementia | 26 | 6.3 |
| Cancer | 26 | 6.3 |
| Worse absorption of vitamins | 8 | 2.0 |
| Other reasons | 33 | 8.0 |

n,%: absolute and relative frequency

**Table 4. Multivariate model by logistic regression of factors associated with long-term use of PPIs (>3 years).**

| Characteristics | Univariate analysis | | | | Multivariate analysis | | | |
|---|---|---|---|---|---|---|---|---|
| | OR | IC95% | | p | OR | IC95% | | p |
| Age in Years | 1.034 | 1.020 | 1.048 | <0.001 | 1.032 | 1.017 | 1.046 | <0.001 |
| Female | 0.931 | 0.596 | 1.453 | 0.752 | - | - | - | - |
| Smoker | 1.171 | 0.468 | 2.926 | 0.736 | - | - | - | - |
| Alcohol use | 1.388 | 0.752 | 2.563 | 0.294 | - | - | - | - |
| BMI | 1.073 | 1.024 | 1.125 | 0.003 | 1.070 | 1.017 | 1.124 | 0.008 |
| No other health problems | 0.478 | 0.292 | 0.781 | 0.003 | - | - | - | - |
| Heart problems | 2.001 | 1.289 | 3.107 | 0.002 | - | - | - | - |
| Bone problems | 1.037 | 0.421 | 2.554 | 0.937 | - | - | - | - |
| Do not use other medicines | 0.429 | 0.256 | 0.718 | 0.001 | - | - | - | - |
| ATC class | | | | | | | | |
| Angiotensin II receptor blockers | 1.286 | 0.780 | 2.119 | 0.324 | - | - | - | - |
| HMG CoA reductase inhibitors | 2.735 | 1.642 | 4.552 | 0.000 | - | - | - | - |
| Biguanides | 1.439 | 0.778 | 2.662 | 0.246 | - | - | - | - |
| Thiazides | 1.040 | 0.538 | 2.011 | 0.908 | - | - | - | - |
| Multivitamins with minerals | 1.872 | 0.973 | 3.602 | 0.060 | - | - | - | - |
| Beta blocking agents | 1.204 | 0.588 | 2.469 | 0.611 | - | - | - | - |
| Thyroid hormones | 1.434 | 0.706 | 2.912 | 0.319 | - | - | - | - |
| ACE inhibitors | 1.599 | 0.736 | 3.477 | 0.236 | - | - | - | - |
| Heparin group | 1.631 | 0.766 | 3.471 | 0.205 | - | - | - | - |
| Benzodiazepine derivatives | 1.540 | 0.660 | 3.593 | 0.318 | - | - | - | - |
| Selective serotonin reuptake inhibitors | 3.853 | 1.577 | 9.418 | 0.003 | 3.522 | 1.397 | 8.882 | 0.008 |
| Platelet aggregation inhibitors excl. heparin | 3.497 | 1.408 | 8.685 | 0.007 | 3.188 | 1.205 | 8.436 | 0.020 |

[11]. This increase in the use of PPIs via self-medication can be explained by its low cost and the great variability of brands, as highlighted by Bomba et al., in a Spanish study that for 6 years followed the prescriptions of PPIs in the province of Araba through retrospective analysis [12]. We observed a lower occurrence of self-medication in our sample. However, the studies cited above are based on retrospective analysis of prescriptions in unrepresentative samples. Our data were collected prospectively from a sample of pharmacy users in the community, proportionally covering a population of about 800,000 inhabitants.

Similar to our results, an Italian study in nine pharmacies with 260 users and also carried out through structured interviews, pointed out polypharmacy and preventive gastroprotection as the main unlicensed indications for PPI. The authors observed that 30% of users are unaware of the duration of treatment with PPI [13], a characteristic also identified in our data. A systematic review of cohort and case-control studies reported the increase in the prolonged use of PPIs being a result of unnecessary prescriptions for "inadequate conditions or unlicensed indications" [14]. Despite this widespread use for several purposes, we observed proper follow-up of the PPI dosage and correct administration.

One aspect that could hinder therapeutic adherence would be the recommendation for fasting before use. However, most users follow this recommendation, and usage is predominant about 30 minutes before breakfast. The most worrying aspect of the PPI usage profile refers to the significant portion of individuals using it for a long time.

The multivariate model indicated a relationship between long-term use and old age, BMI, use of NSAIDs and antidepressants. Elderly and obese individuals have a higher occurrence of gastroesophageal reflux and other dispeptic conditions [15], justifying the greater use of PPIs in individuals with these characteristics. Dyspepsia is a relevant and common adverse reaction in users of NSAIDs and selective serotonin reuptake inhibitors [16], so the administration of PPIs to prevent or treat gastrointestinal discomfort would be expected in users of these drugs. In addition, gastrointestinal discomfort resulting from the use of aspirin for cardiovascular prevention is a frequent indication for PPIs [17]. As far as we know, there are no studies in the literature that use a multivariate approach to identify factors associated with prolonged use of PPI. However, despite the abundance of risk studies resulting from the long-term use of PPIs, the vast majority are observational and the quality of evidence is consistently low or very low [14, 18]. Considering the incidence per patient / year, these potential problems vary between 0.1 to 1%, except for the occurrence of enteric infections which range from 3 to 16% patients / year [17].

Despite a relatively low incidence, the long-term use of PPI can cause negative outcomes for the patient, especially in those with the risk factors identified in our study. Community pharmacists are effective in identifying adverse events in dyspeptic patients, in addition to promoting better adherence to treatment and changes in lifestyle [19]. In this context, we highlight the potential of the community pharmacist to identify the inappropriate use of PPI and its complications [20].

This study had some limitations. It was restricted to only one urban area and a single pharmacy chain. We did not have access to a clinical history to confirm the self-reported diagnosis or when there was a report of a prescribed drug without the presence of a prescription, therefore there is a possibility of memory bias. For future perspectives, the importance of investigating the potential damage involved in the prolonged use of PPI is highlighted. As well as this, there should be studies carried out on the use of medicines in other OTC products.

## Conclusion

Pantoprazole and Omeprazole are the most purchased PPIs by the population of the city of Natal in Brazil. There were no significant deviations regarding the indications for use and

administration that would raise major concerns regarding the safety of the sale of PPIs. However, it can be used for a prolonged period of time, which is characteristic of older patients, those who use anti-inflammatory and antidepressant drugs and those who are overweight. Users are unaware of the potential risks of these drugs, moreover, good quality evidence that characterizes the adverse effects related to the prolonged use of PPIs is not well understood in the literature.

## Supporting information

**S1 Data.**
(XLSX)

## Acknowledgments

The authors acknowledge the essential participation of the students who contributed to the research Renata Avelino, Fernanda Figueiredo, Júlia Scarlet and Heloisa Silva.

## Author Contributions

**Conceptualization:** Lorena Maria Lima de Araújo, Rand Randall Martins, Antonio Gouveia Oliveira.

**Data curation:** Lorena Maria Lima de Araújo, Maria Vivyanne de Moura Lopes, Rafael Silva de Arruda.

**Formal analysis:** Rand Randall Martins, Antonio Gouveia Oliveira.

**Funding acquisition:** Rand Randall Martins, Antonio Gouveia Oliveira.

**Investigation:** Lorena Maria Lima de Araújo, Maria Vivyanne de Moura Lopes, Rafael Silva de Arruda.

**Methodology:** Maria Vivyanne de Moura Lopes, Rafael Silva de Arruda, Rand Randall Martins, Antonio Gouveia Oliveira.

**Project administration:** Lorena Maria Lima de Araújo, Rand Randall Martins, Antonio Gouveia Oliveira.

**Resources:** Maria Vivyanne de Moura Lopes, Rafael Silva de Arruda, Rand Randall Martins.

**Supervision:** Lorena Maria Lima de Araújo, Rand Randall Martins, Antonio Gouveia Oliveira.

**Validation:** Antonio Gouveia Oliveira.

**Writing – original draft:** Lorena Maria Lima de Araújo, Rand Randall Martins, Antonio Gouveia Oliveira.

**Writing – review & editing:** Lorena Maria Lima de Araújo, Rand Randall Martins, Antonio Gouveia Oliveira.

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
