## [Decision Letter · Decision Letter 0]

17 Mar 2021

PONE-D-21-04933

Proton pump inhibitor and community pharmacies: usage profile and factors associated with long-term use

PLOS ONE

Dear Dr. Martins,

Thank you for submitting your manuscript to PLOS ONE. After careful consideration, we feel that it has merit but does not fully meet PLOS ONE’s publication criteria as it currently stands. Therefore, we invite you to submit a revised version of the manuscript that addresses the points raised during the review process.

We look forward to receiving your revised manuscript.

Kind regards,

Sanjiv Mahadeva, MRCP, MD

Academic Editor

PLOS ONE

Journal Requirements:

'Yes. This study was financed in part by

the Coordenação de Aperfeiçoamento de Pessoal de Nível Superior - Brasil (CAPES) - Finance Code 001.'

Additional Editor Comments (if provided):

The study lacks novelty, but is reasonably well presented. The authors need to respond to the Reviewer's comments.

Reviewers' comments:

Reviewer's Responses to Questions

**Comments to the Author**

1. Is the manuscript technically sound, and do the data support the conclusions?

Reviewer #1: No

Reviewer #2: Yes

2. Has the statistical analysis been performed appropriately and rigorously? 

Reviewer #1: No

Reviewer #2: Yes

3. Have the authors made all data underlying the findings in their manuscript fully available?

Reviewer #1: Yes

Reviewer #2: Yes

4. Is the manuscript presented in an intelligible fashion and written in standard English?

Reviewer #1: Yes

Reviewer #2: Yes

5. Review Comments to the Author

Reviewer #1: Thank you for the submission. However, I believe that the manuscript does not have sufficient international interest, novelty and scientific rigour for publication in PLOS ONE. It would add nothing new to the literature. There have been dozens of better studies examining long-term PPI use. There is an implication here in the introduction section that the study is solely examining the OTC use of PPIs without prescription, but that does not seem to be the case at all.

How many patients declined to complete the survey, of those approached? Details of the statistical methods are inadequate e.g. How was the issue of collinearity/multicollinearity in the multivariate statistical analyses dealt with (assessed and appropriately managed)?

The assessment of appropriateness of long-term therapy is also markedly inadequate. Judging against licensed indications is certainly not sufficient. There should be assessment against relevant international or national guidelines specific to the indications accepted as clinically appropriate for the long-term use of these drugs (many such guidelines exist).

Reviewer #2: Thank you for inviting me to review this manuscript which describes the use of PPI in community pharmacy, and the factors associated with it.

Overall, the manuscript is well written. However, I have a few comments that could perhaps improve the quality of the manuscript.

Methods

Please provide more details on how participants were recruited. The authors did not mention that it was only from one pharmacy chain on lines 90-94, but later mentioned this as a limitation that only one chain pharmacy was approached. Also, it was not explained how the four districts in the city were represented, since there will be international readers who will be reading this manuscript. How was the inclusion of the number of participants ensured that it was approximately proportional to the population of each district?

Data analysis

Was data normally distributed? if no, then non parametric tests should be used.

Results

Table 1: what does 0-5 salaries mean?

Also, the column of the table should specify that it is n and %. For continuous variables, it should be presented as 54.1+/- 17.9, rather than in columns. 95% CI values not need for categorical variables. only for the multiple logistic regression table. Also for self reported diseases, suggest to just report the top 3 diseases, and medication ATC class

Line 126: Should avoid using the word "stand out"

Discussion

Should include a para on the long term effects of PPI, and the role of community pharmacists in this?

6. PLOS authors have the option to publish the peer review history of their article (what does this mean?). If published, this will include your full peer review and any attached files.

Reviewer #1: No

Reviewer #2: **Yes: **Pauline Siew Mei Lai

---

## [Author Response · Author response to Decision Letter 0]

28 Apr 2021

Journal Requirements

Response:

Document readjusted according to the model.

'Yes. This study was financed in part by the Coordenação de Aperfeiçoamento de Pessoal de Nível Superior - Brasil (CAPES) - Finance Code 001.'

Response:

Funding Statement corrected, added to the cover letter.

Reviewer #1:

Thank you for the submission. However, I believe that the manuscript does not have sufficient international interest, novelty and scientific rigour for publication in PLOS ONE. It would add nothing new to the literature. There have been dozens of better studies examining long-term PPI use.

1. There is an implication here in the introduction section that the study is solely examining the OTC use of PPIs without prescription, but that does not seem to be the case at all.

Response: 

We are well aware of the large number of reports in the literature on inadequate prescription and inadequado use of PPI's, but all those studies have been conducted in hospital or clinic settings. Those studies have consistently found high rates of physician prescription errors and excessive treatment duration. Given that information, it would be natural to inquire whether those problems are even more worrisome when PII's are dispensed as OTC. Surprisingly enough, after a thourough literature search we found no study looking into the topic of inadequate use of PPI's diapensed OTC, and for that matter of any other OTC drug. Therefore, we conducted the first ever drug utilization study, to the best of our knowledge, of an OTC drug and, ibterestingly, our results have not shown that PPI's dispensed as OTC are a significant risk to patients. Our results address a knowledge gap regarding utilization of OTC drugs.

2. How many patients declined to complete the survey, of those approached? Details of the statistical methods are inadequate e.g. How was the issue of collinearity/multicollinearity in the multivariate statistical analyses dealt with (assessed and appropriately managed)? 

Response: 

As suggested, we added information on the proportion of patients who refused to participate (page 6, lines 134-6). 

“During the execution of the study, about 520 patients were invited and approximately 20% refused to answer the questionnaire, the main reason for refusal was short time.”

Additionally, regarding the assessment of multicollinearity, we added this analysis as suggested (page 6, lines 129-32). 

“Multicollinearity was tested by calculating variance inflation factors (VIF) for all explanatory variables in the full and the minimal multivariate model. The full model had all values of VIF < 4.5 and the minimal model had all values of VIF< 2.0, and hence no problems were displayed.”

We did not identify any problems in adjusting the model as highlighted below. Analysis performed on stata (vif command), after multiple linear regression using age as a dependent variable.

 Variable | VIF 1/VIF 

+----------------------------

Long-term use 1.07 0.934115

Selective serotonin reuptake inhibitors 1.03 0.968357

Platelet aggregation inhibitors excl. heparin 1.03 0.972665

BMI 1.03 0.974237

+------------------------------

Mean VIF 1.04

3. The assessment of appropriateness of long-term therapy is also markedly inadequate. Judging against licensed indications is certainly not sufficient. There should be assessment against relevant international or national guidelines specific to the indications accepted as clinically appropriate for the long-term use of these drugs (many such guidelines exist).

Response:

Position statements or guidelines have provided comprehensive and rational clinical advice concerning long-term use but have not provided a clear definition of what long-term use is. The threshold for defining long-term PPI use varied from >2 weeks to >7 years of PPI use (FREEDBERG, 2017). Among the various risks associated with the prolonged use of PPI, we focus on those related to the occurrence of peptic ulcer and gastroesophageal reflux disease. In these cases, the use for more than 3 years implies an increase in serum gastrin levels and a higher occurrence of gastritis with a predominance of bodies with atrophy, a known risk factor for the development of gastric cancer (LUNDELL et al, 2015). We have added the highlighted texts below (page 6, lines 118-21):

“Individuals with 3 or more years of frequent use of PPIs were considered to demonstrate prolonged use. This time of use is associated with the appearance of changes in gastrin levels and gastric histopathology, initial factors for the occurrence of several complications of long-term use (8).”

Freedberg DE, Kim LS, Yang Y-X. The risks and benefits of long-term use of proton pump inhibitors: expert review and best practice advice from the American gastroenterological association. Gastroenterology 2017;152:70615. 

8. Lundell L, Vieth M, Gibson F, et al. Systematic review: the effects of long-term proton pump inhibitor use on serum gastrin levels and gastric histology. Aliment Pharmacol Ther 2015;42:649–63. 

Reviewer #2:

Thank you for inviting me to review this manuscript which describes the use of PPI in community pharmacy, and the factors associated with it. Overall, the manuscript is well written. However, I have a few comments that could perhaps improve the quality of the manuscript.

4. Please provide more details on how participants were recruited. 

Response: 

More details on recruiting participants have been added. The previous text was:

“All individuals who acquired PPI were approached about their interest in answering the questionnaire.”

Changed to (page 5, lines 102-9):

“In each pharmacy, a consecutive sample of pharmacy users was made. All individuals who acquired PPI were approached about their interest in answering the questionnaire. During the day shift (as 8 a.m. to 4 p.m.), the interviewer remained inside the establishment and approached the participant only after the purchase was completed. The questions were asked orally in a place reserved by the main researcher (pharmacist, LMLA) and auxiliaries (pharmacy students, MVM and RSA). Before the start of data collection, we carried out a pilot study with 10 patients to adjust the questionnaire and train the research team.” 

5. The authors did not mention that it was only from one pharmacy chain on lines 90-94, but later mentioned this as a limitation that only one chain pharmacy was approached. Also, it was not explained how the four districts in the city were represented, since there will be international readers who will be reading this manuscript. How was the inclusion of the number of participants ensured that it was approximately proportional to the population of each district?

Response: 

We have added the highlighted texts below (pages 4-5, lines 89-100):

“The sample size was calculated at 385 participants. This quantity ensures, with 95% confidence, a maximum error of the estimates of ± 5 percentage points. The municipality of Natal, located in the Northeast of Brazil, has 800,000 inhabitants distributed among four health districts (east, south, north and west districts). The community pharmacy chain has 22 establishment throughout the city. Health district is a geographic area that comprises a population with similar epidemiological and social characteristics, in addition to the health resources to serve it (7). In the city evaluated, 38% of the population lives in the northern health district, followed by the western (27%), southern (21%) and eastern (14%) districts.

In order to obtain a sample that represented the different segments of the city's population, the selection and recruitment of participants were conducted in community pharmacies from the 4 districts of the city, with the inclusion of a number approximately proportional to the population of each district. Only one community pharmacy chain participated in the research, consisting of 22 establishments distributed throughout the city (3 pharmacies in the north district, 2 in the west, 11 in the south and 6 in the east).”

7. World Health Organization (WHO). Financial Management: An Overview and Field Guide for District Management [Internet]. Pretoria, South Africa; 2002 [cited 2021 apr 12]. Available from: www.who.int/management/Finances3DistManagement.pdf

6. Was data normally distributed? if no, then non parametric tests should be used.

Response:

The inferential method was multivariate logistic regression (LR) where the dependent variable is binary and of a qualitative nature. One of the prerequisites for using the LR is that it does not depend on assumptions of normality and equality between the variance-covariance matrices (HAIR, 2009). Except for the reviewers' best judgment, we consider an analysis of the normality of the data to be unnecessary.

HAIR, Joseph F. et al. Multivariate data analysis. 6. ed. Porto Alegre: Bookman, 2009.

7. Table 1: what does 0-5 salaries mean?

Response: 

The term was corrected to “minimum wage” and adding the corresponding dollar amount in table 1 footer. 

“In 2019, the Brazilian minimum wage per month is $ 998 Reais and is equivalent to $ 257 US Dollar.” 

8. Also, the column of the table should specify that it is n and %. For continuous variables, it should be presented as 54.1+/- 17.9, rather than in columns. 95% CI values not need for categorical variables. only for the multiple logistic regression table. Also for self reported diseases, suggest to just report the top 3 diseases, and medication ATC class.

Response: 

Corrected tables as suggested.

9. Line 126: Should avoid using the word "stand out"

Response: 

Substituted word (page 9, line 152):

“The occurrence of long-term use (over 5 years) stand out was observed in 27.5% of patients.”

10. Should include a para on the long term effects of PPI, and the role of community pharmacists in this?

Response: 

Paragraph added as suggested (pages 17, lines 236-241):

“Despite a relatively low incidence, the long-term use of PPI can cause negative outcomes for the patient, especially in those with the risk factors identified in our study. Community pharmacists are effective in identifying adverse events in dyspeptic patients, in addition to promoting better adherence to treatment and changes in lifestyle (19). In this context, we highlight the potential of the community pharmacist to identify the inappropriate use of PPI and its complications (20).”

19. Boardman HF, Heeley G. The role of the pharmacist in the selection and use of over-the-counter proton-pump inhibitors. Int J Clin Pharm. 2015;37:709–716.

20. Alhossan A, Alrabiah Z, Alghadeer S, Bablghaith S, Wajid S, Al-Arifi M. Attitude and knowledge of Saudi community pharmacists towards use of proton pump inhibitors. Saudi Pharm J. 2019;27(2):225-228.

---

## [Decision Letter · Decision Letter 1]

24 May 2021

Proton pump inhibitor and community pharmacies: usage profile and factors associated with long-term use

PONE-D-21-04933R1

Dear Dr. Martins,

We’re pleased to inform you that your manuscript has been judged scientifically suitable for publication and will be formally accepted for publication once it meets all outstanding technical requirements.

Kind regards,

Sanjiv Mahadeva, MRCP, MD

Academic Editor

PLOS ONE

Additional Editor Comments (optional):

The authors have responded satisfactorily to the 2 original reviewers comments.

Reviewers' comments:

Reviewer's Responses to Questions

**Comments to the Author**

1. If the authors have adequately addressed your comments raised in a previous round of review and you feel that this manuscript is now acceptable for publication, you may indicate that here to bypass the “Comments to the Author” section, enter your conflict of interest statement in the “Confidential to Editor” section, and submit your "Accept" recommendation.

Reviewer #2: All comments have been addressed

2. Is the manuscript technically sound, and do the data support the conclusions?

Reviewer #2: Yes

3. Has the statistical analysis been performed appropriately and rigorously? 

Reviewer #2: No

4. Have the authors made all data underlying the findings in their manuscript fully available?

Reviewer #2: Yes

5. Is the manuscript presented in an intelligible fashion and written in standard English?

Reviewer #2: Yes

6. Review Comments to the Author

Reviewer #2: Normality of data should be performed regardless, and if found to be not normally distributed, data should be presented as median and interquartile range instead of mean and SD. Otherwise, the authors have addressed all comments appropriately.

7. PLOS authors have the option to publish the peer review history of their article (what does this mean?). If published, this will include your full peer review and any attached files.

Reviewer #2: No

---

## [Editor Report · Acceptance letter]

2 Jun 2021

PONE-D-21-04933R1 

Proton pump inhibitor and community pharmacies: usage profile and factors associated with long-term use. 

Dear Dr. Martins:

I'm pleased to inform you that your manuscript has been deemed suitable for publication in PLOS ONE. Congratulations! Your manuscript is now with our production department. 

Kind regards, 

on behalf of

Dr. Sanjiv Mahadeva 

Academic Editor

PLOS ONE